# Predicting Spin-Dependent Phonon Band Structures of HKUST-1 Using Density Functional Theory and Machine-Learned Interatomic Potentials

**DOI:** 10.3390/ijms25053023

**Published:** 2024-03-05

**Authors:** Nina Strasser, Sandro Wieser, Egbert Zojer

**Affiliations:** Institute of Solid State Physics, NAWI Graz, Graz University of Technology, 8010 Graz, Austria; nina.strasser@tugraz.at (N.S.); sandro.wieser@alumni.tugraz.at (S.W.)

**Keywords:** density functional theory, harmonic lattice vibrations, HKUST-1, machine-learned force fields, metal–organic frameworks, moment tensor potentials, phonons, spin polarization

## Abstract

The present study focuses on the spin-dependent vibrational properties of HKUST-1, a metal–organic framework with potential applications in gas storage and separation. Employing density functional theory (DFT), we explore the consequences of spin couplings in the copper paddle wheels (as the secondary building units of HKUST-1) on the material’s vibrational properties. By systematically screening the impact of the spin state on the phonon bands and densities of states in the various frequency regions, we identify asymmetric -COO- stretching vibrations as being most affected by different types of magnetic couplings. Notably, we also show that the DFT-derived insights can be quantitatively reproduced employing suitably parametrized, state-of-the-art machine-learned classical potentials with root-mean-square deviations from the DFT results between 3 cm^−1^ and 7 cm^−1^. This demonstrates the potential of machine-learned classical force fields for predicting the spin-dependent properties of complex materials, even when explicitly considering spins only for the generation of the reference data used in the force-field parametrization process.

## 1. Introduction

Metal–organic frameworks (MOFs) are a class of versatile materials that consist of metal-oxide nodes connected by organic linkers [1]. The resulting coordination network is typically characterized by a high porosity that allows small guest molecules to be incorporated into the pores [2,3]. The porosity can be useful in catalytic reactions by trapping catalysts inside the pores, preventing them from becoming deactivated by other reactive species, ultimately extending their lifetimes [4]. It also permits the storage [5] or separation of various gases [6]. Moreover, by enclosing active pharmaceutical ingredients inside the pores of MOFs, a delay in the effectiveness of orally taken drugs can be achieved [7]; and by encapsulating proteins into MOF pores, their denaturation can be prevented [8].

Many of the properties of MOFs are determined by phonons, which are the energy quanta of lattice vibrations in 3D periodic materials [9]. Γ-point phonons defined in the centers of the Brillouin zone can be studied, for example, by infrared and Raman spectroscopy. This is, however, not sufficient, as one needs to know phonons in the entire reciprocal space to determine the thermodynamic stability of a material, to model its thermal expansion, or to understand heat and charge-transport properties [10,11]. Notably, for the latter processes, anharmonic phonon properties also become relevant. Additionally, phonons are intimately related to the elastic properties of solid materials, which will become relevant again later, when discussing phonons in the long-wavelength limit.

On more technical grounds, studying phonons and their band structures requires the consideration of periodic boundary conditions, which relies on the periodic structure of crystalline materials. Periodic boundary conditions are well compatible with describing MOFs, which typically display a rather high degree of crystallinity. Methodology wise, when requiring ab initio quality data, this usually calls for the use of (dispersion-corrected) density functional theory (DFT), which—especially when relying on generalized gradient functionals—is particularly well compatible with using periodic boundary conditions.

Material wise, the main focus of this study is on HKUST-1 [12]. It was one of the first synthesized MOFs and consists of Cu_3_(BTC)_2_ units (BTC = 1,3,5-benzene tricarboxylate). The main structural features of HKUST-1 (Figure 1a) are paddle wheels (Figure 1b) comprising two metal centers (Cu^2+^) connected via four carboxylic groups belonging to the BTC linker molecules. According to ligand field theory, these linkers are arranged around the Cu^2+^ nodes in a square-planar conformation due to orbital directing effects [13]. Moreover, BTC is considered to be a weak field ligand that induces a minimal splitting of the d-orbitals as a consequence of its large size and low polarizability [14].

When considering the magnetic properties of HKUST-1, the unpaired electrons residing in the d_x_^2^_−y_^2^ orbitals of the Cu^2+^ ions (which have a d^9^ electronic configuration) can either couple ferro- or antiferromagnetically within each paddle wheel (neglecting the very weak coupling between different paddle wheels, which are rather distant). In the ferromagnetic (FM) state, the spins of the unpaired electrons within a paddle wheel are aligned parallel, giving rise to a triplet configuration. Conversely, in the antiferromagnetic (AFM) arrangement, the spins of the unpaired electrons point in opposite directions resulting in an open-shell singlet structure [15]. Measurements of the magnetic susceptibility [16] as well as spin resonance experiments [17] on HKUST-1 suggest an AFM ground state especially for temperatures below 50 K and 70 K, respectively. Several computational studies [18,19] based on DFT conducted at effectively 0 K also confirmed the AFM ground state for HKUST-1. 

For the non-magnetic (NM) situation generated by switching off the spin polarization in the simulations, the system is further destabilized, because a δ-bond is formed. In that case, both unpaired electrons are forced into the same orbital resulting in a chemical bond between the two Cu^2+^ ions and a closed-shell singlet structure [20]. In practice, especially when employing wave-function based approaches, theoretical studies would frequently treat the copper paddle wheel units of HKUST-1 as triplets (i.e., assuming a FM coupling), because the open-shell singlet structure of the AFM state can suffer from spin contaminations [19]. These spin contaminations occur in open-shell systems, when the wave function is not an eigenfunction of the total spin operator [21]. The situation is typically less challenging, when relying on charge densities rather than on wavefunctions, i.e., when employing DFT. In fact, it has been shown recently that DFT calculations of bulk structures for battery materials do not deteriorate due to spin contaminations. It was found that the relative stabilities of different phases did not change upon correcting for spin contaminations using the approximated spin projection scheme [22].

However, periodic DFT calculations can become computationally highly demanding, especially for very complex systems, because the costs for solving the corresponding Kohn-Sham equations usually scale with O(N^3^), with N being the number of atoms in the system [23]. This cubic scaling typically limits the applicability of DFT to a few hundred atoms per unit cell and to correspondingly smaller numbers, when the calculations need to be performed many times. This is, for example, the case in molecular dynamics (MD) simulations or when predicting anharmonic phonon properties (like phonon lifetimes or mode Grüneisen parameters [24]). Such anharmonic phonon properties are relevant, for example, for thermal expansion processes [25] or for heat transport [26]. Notably, for complex systems like MOFs even the calculation of harmonic phonon properties with DFT can become a sizable challenge (sometimes even computationally infeasible), as the supercells required for properly converged phonon simulations can easily contain hundreds of atoms.

A possible strategy for massively speeding up simulations dealing with quantities related to the structural dynamics of materials is the use of machine-learned potentials (MLPs) [27,28,29,30], which have also been applied to the modelling of MOFs [31,32]. Additionally, MLPs have been applied to successfully predict phonon properties. This has been done, for example, for carbon-based nanosheets [33,34], carbon nitrides [35], biphenylenes [36], fullerenes [37], and certain prototypical MOFs [38,39]. A promising variant of MLPs are moment-tensor potentials (MTPs), which are obtained via linear regression using a set of basis functions. MTPs provide a good compromise between the required computational costs and the desired accuracy and have been successfully incorporated into the MLIP package [40]. A distinct advantage of MTPs is that, although the basis expansion depends on a polynomial, the computational demand scales only linearly with the number of atoms (O(N) scaling) [41].

Our first research objective comprises a thorough investigation of the spin dependence of phonon-related properties in the prototypical MOF HKUST-1. Thus, after confirming the correct description of the spin states, we perform a comprehensive assessment of the impact of the spin state on the phonon bands employing DFT. As the applicability of DFT is strongly limited due to the associated computational cost (see above), we subsequently assess to what extent the spin dependence of the phonon properties can be quantitatively reproduced by suitably parametrized MTPs. For HKUST-1, the resources required for this parametrization are comparable to calculating the harmonic phonon band structure with DFT. Nevertheless, the availability of MTPs becomes particularly relevant, when, for example, calculating phonon-derived properties, like thermal expansion or heat transport via (non) equilibrium molecular dynamics simulations, or by employing the Boltzmann transport equation, for which anharmonic phonon properties need to be calculated. For materials like HKUST-1, to date, such simulations are clearly impossible using ab initio calculated potential energy surfaces, but become feasible using, e.g., MTPs.

Further, we note that, in a recent study, magnetic moment tensor potentials (mMTPs) have been presented, in which the free parameters in the descriptor are expanded in spin basis functions. These mMTPs were able to predict differences in the phonon dispersion relations of bcc iron in the ferro- and paramagnetic states [42], but to the best of our knowledge, mMTPs have not yet been applied to structures that consist of more than one atomic species. This raises a pertinent query: Can conventional MTPs, which do not contain spin basis functions, reproduce the nuanced variations in phonon bands occurring for distinct spin configurations of HKUST-1 and can this be achieved by merely training separate MTPs on reference data calculated for different spin configurations using DFT? To address this second research question of the current manuscript, different MTPs were trained on sets of approximately 500 spin-dependent reference configurations obtained from active-learning molecular dynamics (MD) trajectories. Subsequently, they were evaluated in terms of their ability to predict spin-dependent phonon band structures.

## 2. Results and Discussion

### 2.1. DFT-Calculated Energetics and Spin Densities of HKUST-1 in the Non-, Ferro- and Antiferromagnetic States

Optimizing the geometries of HKUST-1 for different spin configurations yields essentially identical bond lengths and bond angles, as reported in Appendix A in the Appendix A. This also applies to the NM state despite the associated vanishing spin density. The largely identical geometries of the three spin states result in equivalent diffractograms (see Appendix A in the Appendix A), such that scattering experiments are not able to distinguish between the different spin conformations.

Still, we observe distinct differences in the total energies of the AFM, FM and NM spin configurations of the paddle wheels due to differences in the respective exchange interactions. This can be quantified using the exchange coupling constant, J. For HKUST-1, we define it as minus the energy difference per Cu ion in the paddle wheel between the FM and AFM states [43]
(1)$$-J=\frac{1}{2}\times (E_{FM}-E_{AFM})$$

Following this definition, a negative value of J means that the ground state of the system is antiferromagnetic. E_FM_ and E_AFM_ are obtained from the respective energies per unit cell divided by six to account for the six paddle wheels in each unit cell. J is calculated to be −215 cm^−1^ (see Table 1), when using the VASP code [44] in combination with the Perdew–Burke–Ernzerhof (PBE) functional [45] and employing the settings described in Section 3. This means that the AFM ground state is 53 meV per paddle wheel lower than the FM state. This corresponds to a value of 320 meV per unit cell. Fitting experimental electron spin resonance intensities measured at different temperatures using the Bleaney–Bowers equation [46], yielded a value of J = −185 cm^−1^ for HKUST-1 (with H_2_O molecules axially bonded to the Cu ions) [17]. This is only 14% smaller than the calculated value, which is quite satisfactory, considering that in our simulations no axial ligands were included. These are, however, known to somewhat change the value of J [47]. As another idealization, we assumed a perfectly periodic crystal, while in real-world samples defects are inevitable.

To better understand the nature of the DFT-calculated open-shell singlet, the spin densities of the FM and AFM structures were calculated as the difference between the spin-up and spin-down charge densities [48]. Corresponding isovalue plots at ±0.002 e^−^/Å^3^ are shown in Figure 2. For both states one observes significant spin densities on the Cu atoms, with a non-negligible amount of spin density delocalized onto the neighboring O atoms. Most importantly, the spin densities for the FM state and for the AFM state look very similar apart from the fact that in the AFM state the sign of the spin densities flips between the two halves of the paddle wheels. This strongly suggest that the calculated open-shell singlet is a sensible representation of the AFM state. The only other difference between the FM and AFM spin densities is that for the chosen isovalue, only in the FM case a small amount of spin density is observed on the C atoms of the benzene rings, which point towards the paddle wheels. To also quantitatively show that the calculated states reflect the situation of two unpaired electrons with equal or opposite spins on the two halves of the paddle wheels, we estimated the local magnetic moments projected onto individual atoms.

To obtain these local magnetic moments, the spin-dependent Kohn-Sham states are projected onto a localized basis within the PAW sphere [49]. For the FM state, the magnetic moment projected on the Cu atom amounts to 0.55 µ_B_ and when considering the two Cu atoms and adding the contributions on the eight O and twelve C atoms, one obtains a value of 1.9 µ_B_ per paddle wheel (11.6 µ_B_ for the 6 paddle wheels per unit cell). This is extremely close to the ideally expected total magnetizations of 2 µ_B_ (and 12 μ_B_, respectively). The minor discrepancy is attributed to the projection, which (in analogy to any charge partitioning scheme) can only be approximate. The finding that the total magnetic moment of the AFM state equals 0 μ_B_ is expected, independent of the quality of the description of the open-shell singlet. However, the observation that the absolute values of the projected magnetic moments on the Cu and O atoms in the AFM state are very close to those in the FM state again shows that the chosen methodology provides an accurate description of the open-shell singlet configuration.

### 2.2. Impact of the Spin States of HKUST-1 on Phonon Properties

The essentially equivalent geometries for the three considered spin configurations raise the question, whether other observables related to the (dynamical) structure of HKUST-1 would be more affected by the spin. Such observables could, for example, be phonon properties. To address this question, Γ-point phonon frequencies, phonon band structures, and phonon densities of states (DOSs) were calculated for the NM, FM, and AFM states of HKUST-1 within the harmonic approximation. At a later stage (in Section 2.3) the DFT/PBE-calculated quantities will be compared to the ones calculated with suitably parametrized MTPs. To eventually allow a direct and reliable comparison between DFT/PBE- and MTP-calculated results, we decided to display the DFT/PBE and MTP data next to each other in several of the following graphs. In such combined plots, only the panels denoted as PBE are relevant for the discussion in the current section. The other data will become relevant only for the comparison in Section 2.3.

As a first step, the phonon bands in the low-frequency region (here up to 3 THz ≈ 100 cm^−1^) are compared. This is useful, as low-frequency phonons are most relevant for thermal processes. In the context of heat transport [50], the lowest-frequency phonons are typically characterized by higher phonon group velocities and higher phonon lifetimes. Moreover, their higher thermal occupations (following the Bose-Einstein statistics) also lend them a larger weight. The thermal occupation of phonon modes is also relevant for properties like thermal expansion coefficients, electron-phonon coupling constants, and the heat capacity (especially at low temperatures). Finally, the group velocities of the acoustic phonons close to the Γ-point correspond to the speed of sound and via the Christoffel equation [51] are intimately related to the elastic properties of a material.

The DFT-calculated, low-frequency phonon bands in the high-symmetry directions of HKUST-1 are shown in Figure 3a for the three spin configurations. Interestingly, one observes that certain bands are nearly identical for the different spin conformations, while others display non-negligible shifts. To assess this in more detail, we extracted the shifts of all bands in the center of the 1st Brillouin zone (i.e., at the Γ-point; see Figure 3b). Additionally, we analyzed the band structures in terms of their acoustic vs. optical and transverse vs. longitudinal characters, as shown in Figure 4 for the NM state. This helps to unambiguously identify acoustic modes despite avoided crossings and to correctly distinguish between longitudinal and transverse bands. Technically, the degree of the acoustic character of each eigenmode is determined by the degree to which the corresponding displacements of all atoms in the unit cell have the same magnitude and occur in the same direction. Conversely, the longitudinality of a phonon mode measures to which degree the phonon wave vector and the eigenmode displacements are parallel to each other. Further mathematical details are provided in Section 3.

A comparison between Figure 3a and Figure 4a reveals that the acoustic modes display an only very weak dependence on the spin-state throughout the entire 1st Brillouin zone. Even the deviations in the region of the avoided crossing in the Γ-L direction at ~1.2 THz primarily affect states with a comparably weak acoustic character. It is only in the Γ-X direction at rather large wave vectors that somewhat more pronounced deviations occur for the lower transverse band of the NM state. To further quantify the similarity of the acoustic bands, the group velocities, v_g_, along the three reciprocal high symmetry directions were calculated close to the long wavelength limit as described in in Section 3. They essentially correspond to the sound velocities and are summarized in Table 3. The highest values of v_g_ are consistently found for longitudinal acoustic modes in the Γ-L direction. They amount to approximately 60 THzÅ (corresponding to 6000 m/s) and their values vary only very little between the different spin configurations. Only for the lower transverse acoustic phonon band in the Γ-K direction were there clearly smaller values of v_g_ found for the FM and AFM configurations than in the NM case (they are lower by 12.4% and by 9.8%, respectively)

Considering that typically the acoustic phonons display the highest phonon lifetimes [53], especially for long wavelengths, the rather small magnitudes of the shifts in the acoustic phonon bands suggest that for modelling processes like heat transport, even the computationally much cheaper to obtain NM solution could be sufficient. Testing that assessment in detail, however, goes beyond the scope of the current manuscript.

As far as the low-frequency optical phonons are concerned, Figure 3b illustrates for the Γ-point vibrations that the positions of some bands are hardly affected by the spin-conformation of the paddle wheels, while others experience a non-negligible shift. An unambiguous characterization of the different modes, e.g., in terms of stretching, bending, etc., vibrations of individual atom types, is essentially impossible due to the complex structure of the material resulting in the simultaneous significant displacement of many atoms for each mode. Still, some conclusions can be drawn. To facilitate the discussion, equivalent modes (identified on the basis of the dot products of the eigenvectors [52]) are connected by dashed lines, and these lines are color coded based on the degree of change in frequency between all three spin states. If that change is less than 0.1 THz, the vibration is considered not to be affected by the spin configuration and the connecting line is colored in grey. We identify five such (groups of) vibrations, at approximately 0.56 THz, 1.23 THz, 2.11 THz, 2.44 THz and 2.58 THz (where explicit frequency values refer to the situation in the AFM state). These vibrations can be best characterized as wavy motions of the paddle wheels and BTC linkers, as visualized, e.g., the animations of representative modes provided as Appendix A. Appendix A in the Appendix A contains a listing of the Γ-point frequencies for all modes (including those at similar frequencies) and for all spin configurations. As a second category, two (groups of) vibrations appear at virtually identical frequencies for the AFM and FM spin configurations but undergo a significant shift as soon as spin polarization is no longer considered (i.e., for the NM configuration). They are colored in purple. The first of these modes (at 0.86 THz) involves a framework-wide wavy motion in combination with a paddle wheel rotation. These paddle wheel rotations then dominate for the group of vibrations between 1.34 THz and 1.39 THz. A third category of vibrations undergoes substantial frequency shifts between all three magnetic configurations. These vibrations involve motions of oxygen atoms in an out-of-plane fashion, distorting the paddle wheels from their typical square-planar arrangement. For such vibrations, the connecting dashed lines are highlighted in cyan. In the AFM configuration, they are found between 1.86 THz and 1.90 THz and at 2.77 THz.

To extend the comparison to the entire range of phonons present in HKUST-1 (i.e., considering frequencies up to ~95 THz), a comparison of band structures in a correspondingly extended frequency ranges appears futile. This is because (i) there are too many bands to consider and (ii) the optical bands at higher frequencies are mostly rather flat (as shown for selected frequency regions in the Appendix A, Appendix A). Thus, we resorted to comparing the phonon densities of states (DOSs), which is shown in Figure 5, where the DFT/PBE results are highlighted by blue backgrounds. The first impression when looking at these plots is that the DOSs for the different spin configurations are rather similar, which makes a meaningful identification of differences virtually impossible, especially at the full frequency scale.

Thus, we adopted a different approach to identify frequency regions with a particularly large dependence of phonon frequencies on the spin configuration. It is based on calculating the root mean square differences (RMSDs) of Γ-point phonon frequencies for all pairs of equivalent vibrations at different spin configurations in frequency windows of 10 THz. Equivalent modes are again identified via the dot products of their eigenvectors [52]. The results are shown in Figure 6a–f with orange squares highlighting the comparison between the FM and AFM configurations and blue rectangles highlighting the comparisons with the NM state. Additionally, the situations for the low-frequency region discussed above (0–3 THz) and for the region discussed in more detail in the following (38–50 THz) are illustrated in Figure 6g,h.

Overall, one observes that the deviations between the NM state and the FM and AFM configurations are larger than between the two spin-polarized calculations (FM vs. AFM). In fact, NM vs. FM always yields the largest RMSD values, with the exception of the low-frequency region in Figure 6h, where the RMSD for AFM vs. NM is slightly higher. Notably, of all considered frequency ranges by far the largest RMSD values for Γ-point phonons are found between 40 THz and 50 THz. There, even for the FM vs. AFM comparison an RMSD of approximately 0.30 THz (10 cm^−1^) is obtained and the RMSD value peaks at 0.69 THz (23 cm^−1^) when comparing the NM and the FM states of HKUST-1. This renders the said region particularly suitable for a further assessment of the impact of the spin configuration on the phonons in HKUST-1. In fact, for the following discussion, we will slightly expand the considered region to 38–50 THz such that the chosen frequency range is bordered by band gaps. This has no fundamental impact on the trends in the RMSD values, as is evident from a comparison of Figure 6e,h.

The phonon DOSs for the 38–50 THz region are contained in Figure 7a–c, where at this stage again the blue-shaded parts of the plots are relevant, as they represent the DFT/PBE-calculated data. The DOSs contain contributions from phonons in the entire first Brillouin zone, with most bands being rather flat and some displaying band widths of a few tenth of a THz, as shown in Appendix A in the Appendix A. To ease the identification of trends, Figure 7c shows the spin-dependent shifts of the Γ-point vibrations with equivalent modes connected by dashed lines. Data for the remaining frequency regions are provided in Appendix A (Appendix A) in the Appendix A. Equivalent modes have again been identified via the dot products of the eigenvectors [52]. Infrared (IR) active modes, whose nature will be specifically discussed in the following paragraph, are highlighted by yellow bars.

For the NM and AFM configurations, the high-frequency regions of the plots are dominated by a pronounced single peak at 47.5 THz. As illustrated in panel d, it arises from the superposition of three bands. The near degeneracy of these modes is lifted for the FM state. In particular, the IR-active mode amongst the three modes (an antisymmetric -COO- stretching vibration) shifts to a significantly higher frequency of 48.4 THz. This, together with the significant shifts of additional IR active antisymmetric -COO- stretching modes from 44.6 THz (in the NM configuration) and 46.3 THz (in the AFM configuration) to 47.1 THz (in the FM configuration) results in a rather complex DOS between 47.1 THz and 48.4 THz in the FM case. The IR inactive modes in the region between 44.6 THz and 44.9 THz are rather independent of the spin configuration (with an only minor broadening of the associated frequency range in the NM configuration). The intermediate frequency region is dominated by IR active symmetric -COO- stretching modes, which are essentially spin-independent and are, thus, found in all configurations at approximately 42.6 THz. Likewise, the frequency of the lowest IR active symmetric -COO- stretching mode between 39.8 THz and 39.9 THz is rather insensitive to the spin configuration. As far as the IR-inactive modes in the region between 39.7 THz and 42.7 THz are concerned, certain modes remain unaffected by the spin, while most modes are shifted to higher energies in the FM compared to the NM and AFM cases. Overall, this once more results in a rather complex DOS in that frequency region, whose shape distinctly depends on the spin configuration, especially above 39.8 THz. Further, it is noted that a comparisons of spin-dependent IR spectra (including the frequency region most affected by the spin configurations) can be found in Appendix A (impact of the van der Waals correction; Appendix A), Appendix A (impact of using a hybrid functional, Appendix A) and Appendix A (comparison to experiments, Appendix A) of the Appendix A.

### 2.3. Performance of System-Specifically Trained MTPs Compared to DFT Reference Data

In order to generate system-specifically machine-learned force fields for calculating static and dynamic structural properties, a parameterization against (DFT-generated) training data is required. In this study, we trained four different sets of MTPs using training datasets of approximately 500 reference structures of HKUST-1 in the primitive trigonal unit cell employing an active-learning approach [54] as described in Section 3 and as implemented in the VASP code [44]. Three out of the four sets of MTPs were learned against data obtained for specific spin configurations of HKUST-1. In this way, sets of MTPs trained only on NM, on FM, or on AFM DFT calculations were generated. The fourth MTP, referred to as Sp-I denotes a spin-ignorant MTP. It was trained on the entirety of the training data from each of the active-learning runs performed for specific spin configuration. Consequently, this dataset contains approximately 1500 configurations. The Sp-I MTP represents kind of an ‘averaged spin’ approach that allows us to assess the importance of using spin-dependent training data for reproducing a certain spin state. As the parametrization of an MTP is a stochastic process, each set of potentials contained five independently fitted potentials, where the one with the lowest cost function in the parametrization process was used for generating the data discussed in the following (for more details see Section 3).

In order to evaluate the performance of the spin-dependent MTPs, as a first step, they were applied to predict the total energies of the different spin configurations based on the respective DFT optimized HKUST-1 geometries. Of all MTPs, the AFM-MTP predicts the lowest energy, which is fully consistent with the DFT calculations. When using the FM-MTP, the energy increases by 0.35 eV per unit cell between the AFM and FM states, which is in excellent agreement with the DFT-calculated splitting of 0.32 eV. The NM-MTP predicts an even higher energy, which is 1.02 eV per unit cell above the AFM ground state, again in very good agreement with the DFT result of 0.92 eV per unit cell. Further, we note that optimizing the unit cells with the MTPs hardly changes the geometries. Thus, it is not surprising that for full MTP calculations (i.e., optimizing unit cells and calculating energies with the respective MTPs), the energetic splittings hardly change: the FM state then is 0.39 eV per unit cell above the AFM state and for the NM state the splitting remains at 1.02 eV per unit cell.

The next step is to assess, whether the spin-dependently parametrized MTPs are also able to reproduce the nuanced differences in the spin-dependent phonon bands discussed in the previous section. As a first step, we focus on the low-frequency phonon band structures from Figure 3: comparing panels (a) (DFT/PBE calculations) and (c) (MTP calculations) one sees that the key differences between the phonon bands for the different spin configurations are well reproduced by the differently parametrized MTPs. For the acoustic phonons, this is also manifested in similar group velocities in the long wavelength limit in the DFT/PBE and MTP calculations, as reported in Table 3. For the longitudinal phonons, the MTP-calculated values of v_g_ are generally somewhat lower, but the largest deviations reach only 2.5%. For the transverse phonons, the values of v_g_ are sometimes over-, but mostly underestimated, but deviations are still comparably small. To ease the comparison of DFT/PBE and MTP calculations especially at larger wave vectors, Figure 8 provides a direct comparison of the bands for each spin configuration. This shows that for larger wave vectors more pronounced deviations between the DFT/PBE and MTP results occur for the higher-frequency transverse acoustic phonon bands. Deviations are most pronounced close to the K and X points.

Regarding the optical phonons, a comparison of the Γ-point frequencies (provided in Figure 3c,d) shows that again the MTP calculations excellently match the DFT/PBE results. As far as off-Γ optical phonons are concerned, the excellent agreement prevails in most regions of the plots in Figure 8. Once again close to K and X at frequencies between 0.3 THz and 0.8 THz larger deviations prevail, making this the only region of the low-frequency band structure in which the performance of the MTPs is not fully satisfactory.

At higher frequencies, again the situation between 38 THz and 50 THz is of particular interest, as it contains the most relevant spin-dependent frequency shifts. Thus, in Figure 7, the corresponding, spin-dependent phonon DOSs and shifts in Γ-point frequencies are compared for calculations with DFT/PBE (blue shading) and for calculations with the different MTPs (no shading). From this comparison, it is apparent that the MTPs very well reproduce the spin-dependent differences in the shapes of the phonon DOSs, especially regarding the positions as well as numbers of peaks. This is also illustrated by the similarity of the shifts in Γ-point frequencies inferred from panels (d) and (e). Only the vibrational modes found at approximately 46.3 THz in the DFT/PBE calculation of the AFM state are more strongly underestimated by the chosen MTP, which places them at approximately 45.9 THz (see also discussion in Appendix A in the Appendix A). In contrast to the excellent performance of the spin-specifically parametrized MTPs, the spin-ignorant MTP (Sp-I MTP; contained as grey line in the DOS plots) is unable to accurately predict the overall shapes of any of the DOSs. This confirms the notion that a prediction of spin-dependent phonon properties is intimately linked to a spin-dependent parametrization of the used machine-learned potential. Further, it is noted that most phonon bands relevant for the DOSs in Figure 7 are essentially flat, while some display a non-negligible dispersion. As shown in Appendix A, this is typically also reproduced by the MTPs, although certain deviations in band widths prevail on top of the minor errors in the absolute values of the shifts of the phonon bands apparent already from the shifts of the Γ-point phonons shown in Figure 7.

To provide a more quantitative comparison between the DFT/PBE- and MTP-calculated phonon frequencies and to cover the full frequency range, Figure 9 lists the RMSDs between frequencies (wavenumbers) of Γ-point vibrations of HKUST-1 calculated using DFT/PBE (enforcing certain spin configurations) and spin-sensitively parametrized MTPs for frequency windows of 10 THz. Additionally, in Figure 9g,h, the corresponding plots for the low-frequency region and the region between 38 THz and 50 THz are shown. The entries in each column of the heat map matrices are calculated using the same spin configuration in the DFT/PBE calculations, while the entries in each line are based on using the same MTPs. The diagonals of the matrices compare DFT/PBE calculations for a specific spin configuration with calculations using the MTP parametrized for exactly that configuration. Matrix elements highlighted by orange squares contain the comparisons between the FM and AFM configurations, where one of the configurations has been calculated by DFT/PBE, while the other has been simulated with an MTP. Blue rectangles contain comparisons between the NM configuration and either the FM or the AFM states.

Except for a single entry in the 90–100 THz matrix, the diagonal elements always contain the smallest RMSDs for each frequency range. In fact, the values of the diagonal elements are typically very small (below 0.09 THz (3 cm^−1^) in 17 out of 24 cases and even below 0.06 THz (2 cm^−1^) in a total of 11 cases). This further supports the notion that the spin-selectively trained MTPs are well capable of reproducing the spin-dependent variations in the phonon properties of HKUST-1. A slightly larger value of the RMSD is found in the 40–50 THz frequency window for the AFM state. It amounts to 0.22 THz (7.4 cm^−1^) and a similar value of 0.21 THz (7.0 cm^−1^) is obtained when extending the range to 38–50 THz. This is not entirely unexpected in view of the modes at approximately 46 THz being somewhat more seriously underestimated by the MTP for the AFM configuration (see discussion of the comparison for Figure 7 above). Still, the off-diagonal elements in the corresponding region are much larger, ranging between 11.0 cm^−1^ and as much as 23.0 cm^−1^, showing that the AFM-parametrized MTP still provides by far the best description of the AFM state.

As a final step, we calculated the RMSDs of the frequencies of the Γ-point phonons between the differently parametrized MTPs in analogy to the evaluation of the spin-dependent deviations of the DFT/PBE calculations in Figure 6. The results for the MTPs are shown in Figure 10. When disregarding the entries for the spin-ignorant MTP (forth columns and fourth lines of all matrices), a pattern evolves that is reminiscent of that in Figure 6, once more confirming that the MTPs provide trends similar to those obtained by DFT/PBE. Regarding the spin-ignorant force field, it becomes apparent that its results do not resemble any of those of the spin-selective MTPs. For example, in the lowest- as well as in the highest-frequency regions, the RMSDs when comparing that potential to the NM-MTP are lowest. Between 10 THz and 20 THz, its results most strongly resemble those for the AFM-MTP; between 20 THz and 30 THz, it mimics the FM-MTP; and between 30 THz and 40 THz, it again produces results similar to the NM solution. In the most significant region between 40 THz and 50 THz (or, equivalently, between 38 THz and 50 THz), the Sp-I MTP frequencies compare essentially equally well to those of the AFM and NM cases, albeit with relatively large RMSDs that exceed 0.27 THz (9 cm^−1^). In fact, in this frequency region, the RMSDs are largest for all MTP combinations, reaching a value of 22.0 THz when comparing the NM- and the FM-MTPs. This is essentially equal to the 0.69 THz (23 cm^−1^), obtained when making the same comparison based on the DFT/PBE data. These results are once more in line with the assessment that suitably parametrized MTPs largely reproduce the DFT/PBE trends.

## 3. Materials and Methods

### 3.1. Density Functional Theory Calculations

The structure of HKUST-1 containing 156 atoms in a trigonal primitive unit cell was fully optimized with respect to cell parameters and atomic positions in the NM, FM and AFM states employing the projector-augmented wave (PAW) method and using the Vienna ab initio simulation package [44] (VASP, version: 6.3.0). The exchange-correlation energy functional of Perdew–Burke–Ernzerhof [45] (PBE) was used and augmented by Grimme’s dispersion correction [55] with the Becke–Johnson damping [56] function (D3/BJ). We refrained from including an explicit on-site interaction via the DFT + U approach as (i) already the PBE calculations correctly predicted the AFM ground state of HKUST-1 with an exchange coupling constant rather close to experiments. Moreover, (ii) the DFT + U approach suffers from ambiguities arising from the choice of U and from the choice of corrections to the double-counting problem. Instead, we performed selected calculations using a hybrid functional (PBE0 [57]), a strategy that often provides a very favorable description of quasiparticle energies [58] and which has been considered preferable over DFT + U calculations [59]. In fact, in certain materials, the DFT + U approach has been found to deviate further from DFT plus dynamic mean-field theory calculations than pure, semilocal DFT [60].

The PBE0 [57,61] calculations were performed using the FHI-aims code [62] (version: 221103), where convergence could also be achieved efficiently in the hybrid functional calculations (see Appendix A in the Appendix A). As discussed in Appendix A in the Appendix A (see Appendix A), the main consequence of the hybrid functional calculations is a shift of the vibrational DOSs and IR spectra to higher frequencies, which is consistent with the observed smaller unit cell size and a stiffening of the bonds. This apparently improves the agreement between calculated and measured IR spectra in certain spectral regions, as shown in Appendix A in Appendix A. The PBE0 results, however, suffer from the fact that due to the size and complexity of the HKUST-1 unit cell, the default light basis sets of FHI-aims had to be used, which cannot be considered as converged. Therefore, the PBE0 results only serve for a qualitative comparison. For that reason, and because hybrid functionals are not well suited for parametrizing force fields due to the huge associated computational costs, the focus of the main manuscript is on the PBE data.

In Appendix A (Appendix A), the Appendix A also contains an assessment of the impact of different van der Waals approaches (D3/BJ as mentioned above, D4 [63,64,65], and the many-body dispersion approach [66]). Their impact is, however, much smaller than that of the choice of the functional. In particular D3/BJ and D4 yield essentially identical results.

Convergence tests for the k-point sampling and for the cutoff energy suggested a Brillouin zone sampling with a 1 *×* 1 *×* 1 Monkhorst–Pack [67] k-point grid (owing to the large unit cell and the flat electronic bands of the HKUST-1) and a 900 eV cutoff energy. These convergence tests are provided in Appendix A in the Appendix A (see Appendix A). A Gaussian smearing was applied for describing the occupation of the electronic states setting the corresponding width to 0.05 eV. In the VASP calculations, the global precision “Accurate” was used enforcing a particularly tight spacing of the grids when calculating charge densities, augmentation charges, potentials and, real space projectors [68]. These setting ensure DFT reference data with a high accuracy. The threshold for the energy convergence of the self-consistent field cycle was set to 10^−6^ eV for the geometry optimizations and was increased to 10^−8^ eV for the respective frequency calculations (see below).

Spin-polarized DFT was used for simulating the AM and AFM configurations. The magnetization of the unit cell (which is determined by the difference of spin-up and spin-down electron densities of the entire unit cell) was set either to 0∙µ_B_ (for the AFM solution) or to 12∙µ_B_ (for the FM solution). µ_B_ refers to the Bohr magneton. For obtaining the AFM solution, the initial spin moments of the two Cu atoms belonging to one paddle wheel unit were set to an antiparallel arrangement. The coupling of adjacent paddle wheels was chosen such that the distance between Cu atoms on neighboring wheels having the same initial spin was minimized. In the case of the FM coupling, all Cu atoms in the primitive unit cell were initialized with parallel spins.

In order to confirm the final spin state of the open-shell configurations, spin densities were calculated using VASP (version: 6.3.0) [44]. The results were processed with VASPKIT (version: 1.3.5) [69] and plotted using the visualization for electronic and structural analysis (VESTA, version: 3.5.7) program [70].

Phonopy [71,72] (version: 2.11.0) was used for calculating phonon bands. As a first step, it generated structures with a single atom displaced by 0.01 Å. Phonon frequencies were then calculated from the force constants employing the post-processing tool implemented in Phonopy. Supercell convergence tests for the phonon calculations were performed using the MTPs, as illustrated in Appendix A in the Appendix A. They show (independent of the spin state) that already the (comparably large) primitive unit cell of HKUST-1 provides converged results. In this context, Appendix A in the Appendix A shows that essentially identical phonon bands are obtained for a 2 × 2 × 2 supercell and for the primitive unit cell. In passing, we note that performing phonon calculations on 2 × 2 × 2 supercells of HKUST-1 with converged DFT settings rather than with the MTPs is beyond present computational capacities (at least for the used supercomputer).

Smooth curves for phonon DOS plots were obtained by a Lorentzian broadening with a broadening parameter σ of 0.05 THz. For calculating quantities requiring a Brillouin zone integration, a 10 *×* 10 *×* 10 q-mesh was employed. Eigenvector overlaps were calculated via dot product between two modes calculated, e.g., for different spin states or employing different methodologies and were used to assign equivalent modes.

A metric called the acoustic participation ratio (APR) was used to identify the acoustic character of a mode. It measures to what extent all atoms within the unit cell are displaced in the same direction and is defined as [73]
(2)$${APR}_{q,n}=\frac{2}{N(N+1)}\frac{{\left|\sum_{\alpha =1}^{3N}\sum_{\beta \geq \alpha }^{3N}\frac{{(e}_{q,n}^{\alpha }{)}^{†}e_{q,n}^{\beta }}{\sqrt{m_{\alpha }m_{\beta }}}\right|}^{2}}{\sum_{\alpha =1}^{3N}\sum_{\beta \geq \alpha }^{3N}{\left|\frac{{(e}_{q,n}^{\alpha }{)}^{†}e_{q,n}^{\beta }}{\sqrt{m_{\alpha }m_{\beta }}}\right|}^{2}}$$

$e_{q,n}^{\alpha }$ is the eigenvector displacement of atom α for a mode in band n with wave vector **q**. m_a_ is the mass of that atom. The pre-factor $\frac{2}{N\left(N+1\right)}$ (with N being the number of atoms in the unit cell) ensures a normalization of the acoustic participation ratio such that it approaches unity for modes for which all atoms are displaced in the same direction, as long as the number of oscillating atoms is large enough.

The longitudinality, L, measures to what degree the atoms for a phonon with a certain wave vector **q** are displaced in the same direction parallel to **q**. It is defined as [74]:(3)$$L_{q,n}=\left|\frac{1}{N}\sum_{\alpha }^{N}\frac{qe_{q,n}^{\alpha }}{|q|\sqrt{e_{q,n}^{\alpha }e_{q,n}^{\beta }}}\right|$$

A value of 1 for L characterizes a phonon mode that is entirely longitudinally polarized (with atomic displacements in the **q**-direction). In contrast, for fully transversely polarized modes (with atomic displacements perpendicular to **q**), L becomes 0. Further, we note that this assignment is not bijective, i.e., from a value of L equaling 0 one can only conclude that that mode is transverse, if that mode is also fully acoustic. Otherwise, negative and positive contributions in the numerator of equation (3) could simply cancel yielding L = 0 also for non-transverse modes; thus, L = 0 could also be obtained for a longitudinal optical mode.

For obtaining group velocities, we determined the slope of the phonon bands between the wave vector points at 15% and at 20% of the extent of the Brillouin zone. The reason for choosing these rather large values at the ‘boundary’ of the long-wavelength limit is due to the observation that for the NM configuration all three acoustic bands are found at marginally imaginary frequencies of approximately −0.02 THz at the Γ-point. These we could not avoid despite considerable efforts. In fact, they should not cause any problems considering that these modes correspond to rigid translations of the unit cell. Still, they adversely affected the determination of group velocities for that spin configuration close to Γ and also prevented the reliable use of the internal routines of Phonopy.

### 3.2. Generation of Training Data

The individual training sets for HKUST-1 in the NM, FM and AFM states were obtained in the course of an active-learning MD-based approach that is capable of on-the-fly learning a kernel-based potential [54] employing the VASP code [44] (here using version: 6.3.0). The workflow of this machine-learning approach is illustrated in Figure 11a. In the course of the MD run, the kernel-based MLP is built from a basis set derived from local reference configurations (atoms with their surrounding neighborhoods) [75,76,77]. At each time step, the Bayesian error of the prediction of the MLP for the next step is evaluated and if the error surpassed a dynamically adapted threshold, an ab initio calculation is executed. The DFT-calculated configuration is then added to the training dataset, dynamically expanding the basis set of the MLP. This triggers a regular retraining of the potential using the updated training dataset. The active-learning algorithm is terminated, once it reaches the set number of time steps specified by the user. A more extended description of the active-learning approach in VASP can be found in [77]; additionally, the study in [38] provides a very detailed explanation of the procedure sketched above for complex materials like MOFs. It also includes a discussion, of how a significant speedup of the MLP can be achieved utilizing various options in more recent versions of the VASP code (e.g., VASP 6.4.1).

Independent training sets were generated for AFM, FM, and NM spin configurations in the DFT calculations. Each training set contained approximately 500 DFT-calculated configurations that were later used to train the spin-sensible MTPs. The MD simulations ran over 50,000 steps while gradually heating the system from 100 to 500 K employing a Langevin thermostat in the NPT ensemble using a time step of 1 fs [78,79]. The friction coefficient for the atomic degrees of freedom was set to 1 ps^−1^ for all atom types, while the friction coefficient for the lattice degrees of freedom was set to 10 ps^−1^, with a fictitious mass of 10 amu [78]. The remaining settings regarding the ab initio part of the MD calculation were identical to the ones of the geometry relaxation in VASP. All settings related to the active-learning approach were left at the default values. This includes the Bayesian error threshold that was initially set to 0.002 eV/Å and was dynamically adapted during the MD run at different temperatures. This threshold is responsible for determining additional time steps to be performed with DFT.

### 3.3. Training of Moment Tensor Potentials

In a second step, the DFT-calculated configurations form the above active-learning procedure were then used to train MTPs (which could, for example, be efficiently interfaces with widespread MD packages like LAMMPS [80]). A recent comparison of the efficiency and accuracy of MTPs and the kernel-based MLPs generated with VASP for a variety of MOFs can, for example, be found in [38]. The training of the MTP was performed using the machine-learned interatomic potentials (MLIP) package [40]. The corresponding workflow is shown in Figure 11b. First, the parameters of the MTP are initialized randomly. Then, moment tensors up to the desired level of the MTP are constructed whose contractions are used to compute the interaction energy of the system depending on a user-defined cutoff radius. After a linear regression, the loss function built from the energies, stresses and forces in the training dataset is minimized using the BFGS (Broyden–Fletcher–Goldfarb–Shanno) algorithm [81]. This iterative process continued until the differences in the loss function within the previous 50 iterations reached a value smaller than 2·10^−3^. Again, a much more detailed description of the process with a focus of its application to MOFs can be found in [38].

The MTPs were trained at level 22 with 12 radial basis functions, two settings that define the details of their functional form, as described in [41]. The inner cutoff radius was set to the MLIP-determined value (of 0.93 Å for the NM and Sp-I MTPs, to 1.00 Å for the FM MTP, and to 0.99 Å for the AFM MTP). For the outer cutoff radius, we chose the default value of 5 Å. The cutoffs determine the range over which the interactions between atoms are considered. The parameters of the MTPs were obtained by minimizing a cost function comprising the deviations between MTP predicted and DFT-calculated energies, forces on atoms, and stresses on the unit cell. For the energy, force, and stress weights the default values of 1 eV^−1^, 0.1 Å/eV and 0.001 GPa^−1^ were used. HKUST-1 contains four elements (Cu, C, O, and H), which also define the number of specified atomic species in MLIP in the present case. This means that no further separation between atom types (e.g., between differently hybridized carbon atoms) was performed for the present study. It should be noted that a recent study on MOFs suggest that differentiating between atom types has the potential to further improve the agreement between MTP and DFT results [38]. Due to the stochastic nature of the MTP fitting (arising from the random initialization of the fitted parameters), the training run for each spin configuration was carried out five times with different initializations generating five independent MTPs. The MTP with the lowest cost function was chosen for further evaluation. There would also be alternative, much more costly, but potentially less biased approaches for picking the ‘best MTP’. Systematically evaluating such approaches goes beyond the scope of the present manuscript, but two strategies should be discussed briefly: one possibility would be to evaluate the MTPs based on their performance when describing an independently generated validation set of reference structures. In a recent paper, benchmarking the performance of MTPs for five different (non-magnetic) MOFs, we, however, observed that the MTPs with the lowest cost functions always turned out to also be the ones with the best description of the validation set (even though some rather low cost function MTPs displayed a less favorable performance) [38]. Alternatively, one could also pick the ‘best MTP’ based on the performance when describing Γ-point phonons. In fact (as shown in Appendix A; Appendix A), some of the MTPs with higher loss functions do result in somewhat smaller RMSDs between frequencies (wavenumbers) of equivalent Γ-point vibrations calculated with DFT/PBE and a specific MTP. However, calculating Γ-point vibrations with DFT/PBE is not trivial for systems as complex as HKUST-1 and, thus, does not appear suitable for establishing a ‘routine strategy’. Moreover, choosing the MTPs based on minimized RMSDs would create a serious bias as then their choice and the evaluation of their overall performance would be based on the same quantity.

## 4. Conclusions

The current paper shows to what extent the phonon properties in a complex material like HKUST-1 depend on its magnetic state; more precisely, on the spin configuration of the individual Cu paddle wheels that form the secondary building unit of that widespread MOF. To that aim, on the one hand, state-of-the-art, dispersion-corrected DFT calculations were performed, while, on the other hand, it was also assessed to what extent the spin-dependent dynamic properties of HKUST-1 can be described by classical, but system- and spin-specifically machine-learned potentials.

As a first step, we demonstrate that DFT is capable of describing the AFM, FM and NM spin configuration of HKUST-1, confirming the experimental finding that the AFM state is energetically most favorable. The corresponding exchange constant characteristic of the splitting between the FM and AFM configurations is calculated to be −215 cm^−1^, which is very close to the experimentally determined value of −185 cm^−1^. Also, the shapes of the configuration-dependent spin densities and the distribution of the atomic magnetic moments are consistent with the expectations for FM and AFM configurations of the Cu-paddle wheels.

In the low-frequency region (most relevant for processes like thermal expansion or heat transport), the acoustic phonon bands and the corresponding group velocities in the long-wavelength limit display very little dependence on the spin-state of the paddle wheels. For the low-lying optical phonons, a mixed picture emerges, with some of them strongly depending on the magnetic state of HKUST-1, while others being largely unaffected. At higher frequencies, a statistical analysis reveals that especially the region between the band gaps at 38 THz and 50 THz displays a distinct dependence on the magnetic state of the paddle wheels, with the strongest variations predicted for the IR-active asymmetric -COO- stretching vibrations.

As far as system-specifically learned potentials are concerned, different spin-properties are encoded by fitting them against DFT calculations for specific spin configurations performed during active-learning runs. Directly comparing the predictions of these MTPs for phonon band structures and densities of states to the DFT results testifies to an excellent quantitative performance of the spin-sensitively generated MTPs. This is particularly encouraging, as atomic spins and magnetic moments are no explicit parameters playing a role in the functional form of the used MTPs, but are ‘encoded’ only via the reference data used for their parametrization. The excellent agreement between DFT and MTP results is confirmed by an extended statistical analysis of method- and spin-dependent variations in the calculated Γ-point frequencies.

Finally, it should be noted that, even though the calculation of phonon band structures is many orders of magnitude faster when using an MTP instead of DFT, such calculations are not the primary target for the discussed approach. This is because a significant fraction of the gain in computational efficiency is consumed again by the resources needed for the active-learning and for the parametrization of the MTPs. Consequently, the computational resources needed for calculating phonon band structures by DFT and by MTPs (including their parametrization) are of a similar order of magnitude at least for materials like HKUST-1. The exact ratio severely depends on the symmetry of the system that could be exploited in the DFT phonon calculations, the size of the supercell that needs to be considered for obtaining converged phonon bands, and on more technical aspects, like the used computer architecture. In contrast, when calculating, for example, thermal conductivities, multiple MD runs on supercells containing thousands of atoms need to be performed. Our preliminary results show that for simulating heat transport in HKUST-1 using the approach-to-equilibrium molecular dynamics (AEMD) [82], the largest cells that need to be considered for the finite size extrapolation contain around 100.000 atoms. This number further increases, when instead employing non-equilibrium molecular dynamics (NEMD) [83]. Bearing in mind that DFT displays a cubic scaling with the number of atoms, it becomes obvious that trying to perform AEMD or NEMD simulations using DFT would be futile. Here, MTPs benefit from their linear scaling with N. Moreover, in the above-mentioned calculations, millions of time steps are needed to achieve convergence, while the MTP parametrization needs to be performed only once. Similar considerations apply to calculating, for example, phonon lifetimes or thermal expansion processes. For all these tasks, computationally extremely efficient approaches are required. Here, machine-learned potentials will become real game changers and the development of fast potentials with essentially DFT accuracy will pave the way for reliably modelling the dynamic and temperature-related properties of complex materials. The current paper shows that this is also possible when spin-dependent properties need to be considered.

## Figures and Tables

**Figure 1 ijms-25-03023-f001:**
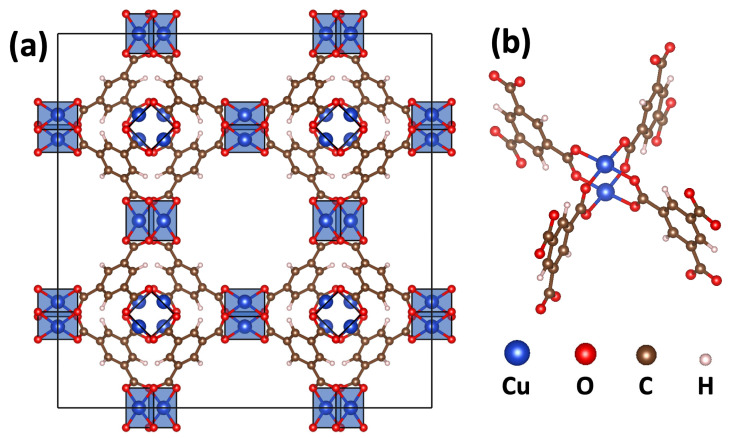
(**a**) Conventional cubic cell of HKUST-1 in space group Fm-3m forming a porous 3D network involving a square-planar coordination geometry around the Cu^2+^ ions as indicated by blue transparent planes. (**b**) Paddle wheel structural motif of HKUST-1.

**Figure 2 ijms-25-03023-f002:**
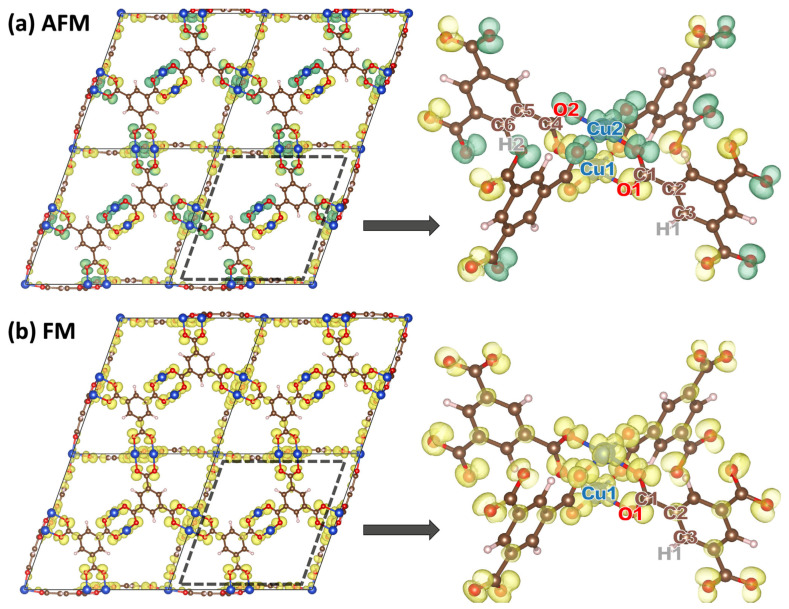
Spin density of AFM HKUST-1 in the 2 × 2 × 2 supercell of the primitive trigonal unit cell. Yellow isosurfaces refer to an excess of spin-up density and green isosurfaces to an excess of spin-down density. The panels to the right show zooms into the region around one paddle wheel, which correspond to the regions indicated by the black dashed lines in the left panels. Alternating signs of the spin densities are found for the AFM case around neighboring Cu atoms comprising one paddle wheel motif (with an interatomic distance of 2.5 Å) (**a**), while the signs are the same for the FM case (**b**). Inequivalent atoms for which local magnetic moments are listed in Table 2 are labeled.

**Figure 3 ijms-25-03023-f003:**
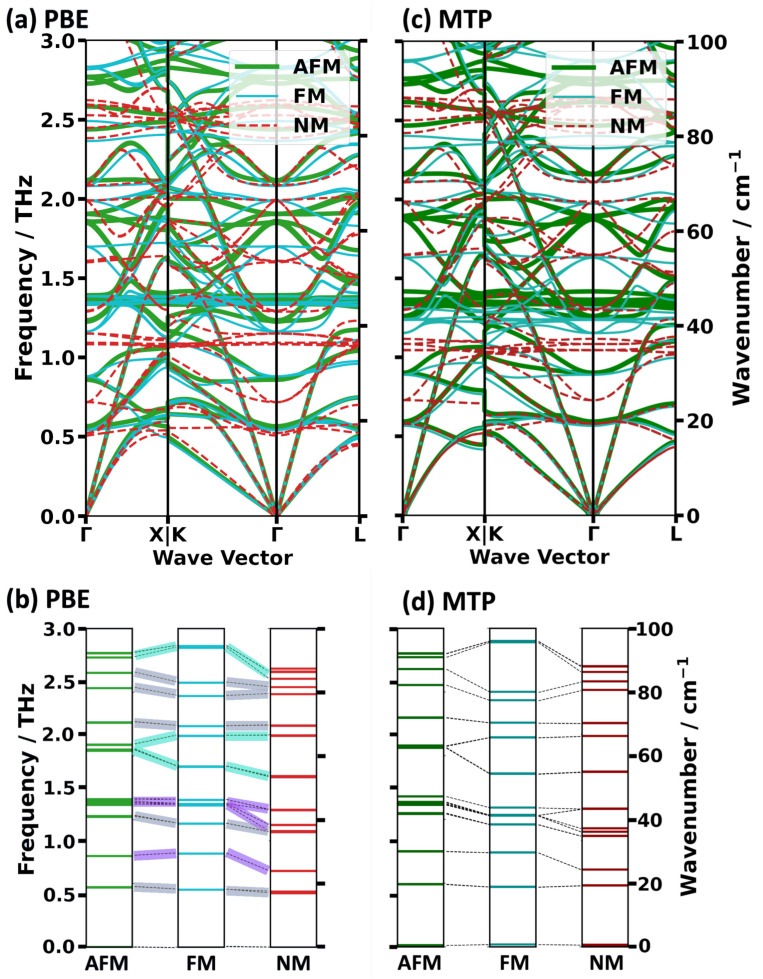
Low-frequency phonon band structures of HKUST-1 in the NM, FM and AFM configurations calculated with DFT/PBE (**a**) and with an MTP parametrized as described in Section 2.3 (**c**). Panels (**b**,**d**) illustrate the shifts between Γ-point frequencies for the different spin states (again calculated with DFT/PBE in panel (**b**) and with suitably parametrized MTPs in panel (**d**)). The frequency and wavenumber scales apply to both panels contained in the same row. In panels (**b**,**d**), modes that have equivalent displacement patterns (identified based on maximum eigenvector dot products [52]) are connected by dashed lines with the respective color code, which depends on the degree of change between the different spin states, described in the main text.

**Figure 4 ijms-25-03023-f004:**
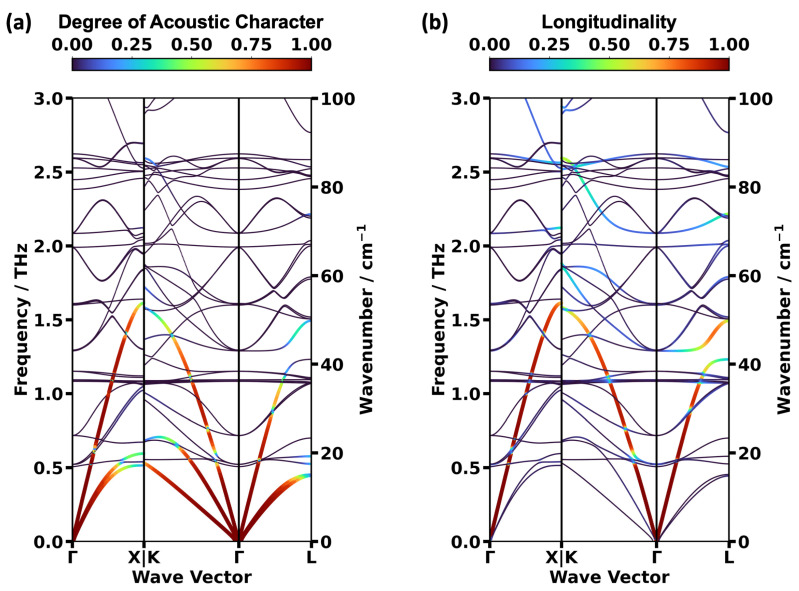
DFT/PBE-calculated low-frequency phonon band structures of HKUST-1 in the NM state colored according to the degree of the acoustic character (**a**) and according to the longitudinality (**b**) of the individual phonon modes (for details see main text and Section 3.

**Figure 5 ijms-25-03023-f005:**
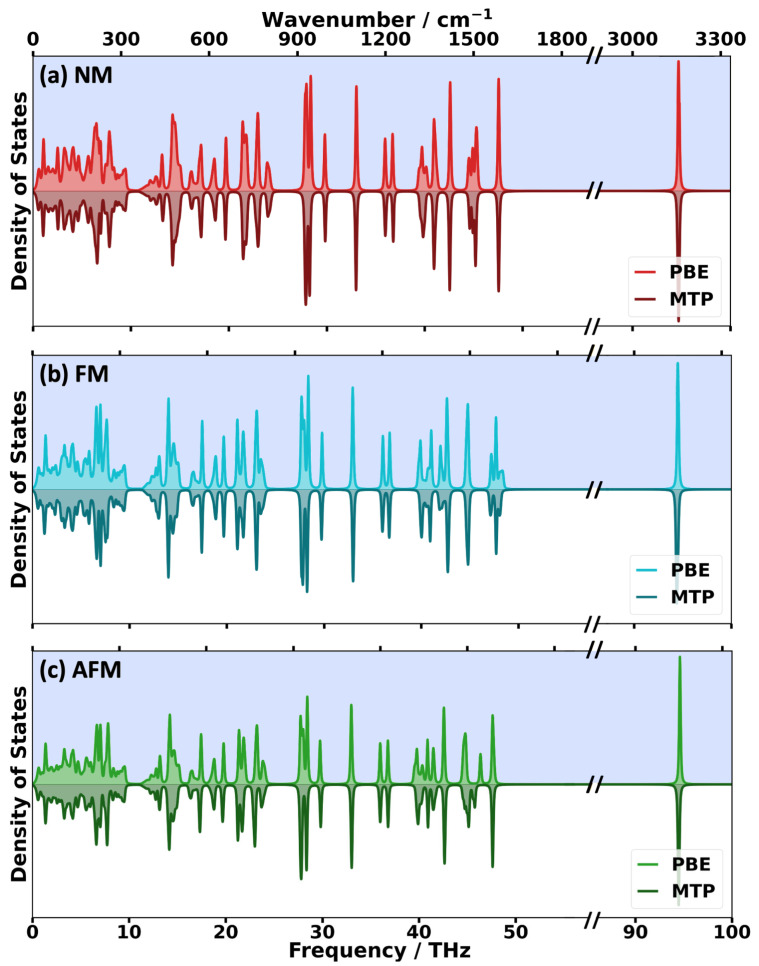
DFT/PBE- and MTP-calculated phonon densities of states (DOSs) for the NM (**a**), FM (**b**) and AFM (**c**) spin configurations for the entire frequency range in which phonons exist (calculated using a 10 × 10 × 10 **q**-mesh). An axis break was introduced, as in HKUST-1, there is an extended phonon band gap between ca. 50 and 90 THz. A broadening parameter, σ, of 0.05 THz was used for the Lorentzian peaks centered at the frequencies of the phonons. 2σ corresponds to the full width at half maximum of the peaks. The DFT/PBE DOSs are plotted with a blue background.

**Figure 6 ijms-25-03023-f006:**
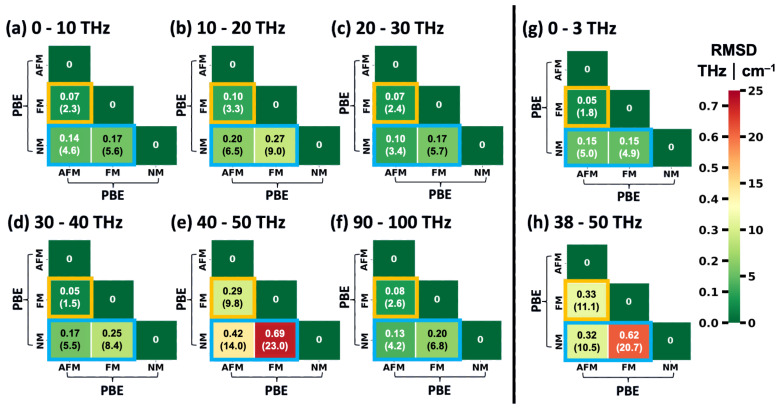
RMSDs between frequencies (wavenumbers) of Γ-point vibrations of HKUST-1 for equivalent modes in different spin configurations. The values are given in THz and also in cm^−1^ in parentheses. They are provided in the form of lower triangular matrices and have been calculated for different frequency regions comprising 10 THz each (panels (**a**–**f**)). Both the columns and rows represent predictions with the PBE functional in VASP for all three spin states. The comparisons between FM and AFM states are highlighted by the orange squares and those between the NM configuration and the two spin-polarized ones by the blue rectangles. Additional plots are shown for the low-frequency region (0–3 THz; panel (**g**)) and for the region discussed in detail in the main text (38–50 THz; panel (**h**)).

**Figure 7 ijms-25-03023-f007:**
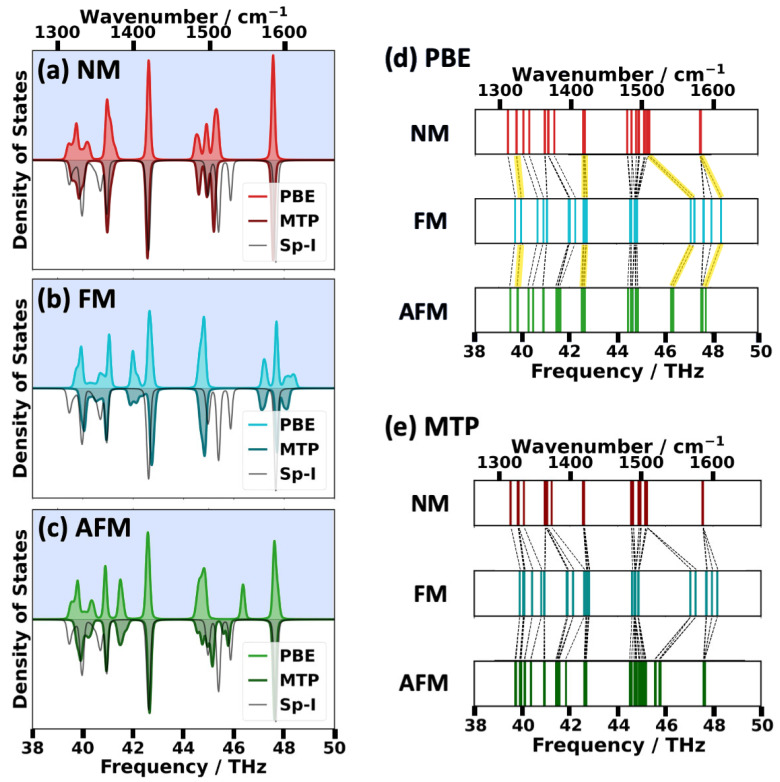
Phonon DOSs for the NM (red) (**a**), FM (blue) (**b**) and AFM (green) (**c**) configurations between 38 THz and 50 THz. The results obtained with DFT/PBE are plotted on the top (with a blue background), while the computations that were performed with the corresponding MTPs are displayed at the bottom in a darker shade of the color. The results for the spin-ignorant (Sp-I) MTP are plotted as a grey line in the MTP regions of all panels. Panels (**d**,**e**) illustrate the shifts as a function of the spin configuration for the frequencies of the Γ-point vibrations in the range between 38 THz and 50 THz calculated with DFT/PBE (**d**) and with the MTPs (**e**). In these plots, vibrations with the highest eigenvector overlaps are connected by grey dashed lines. In panel (**d**), these lines are marked by yellow bars for the IR active modes.

**Figure 8 ijms-25-03023-f008:**
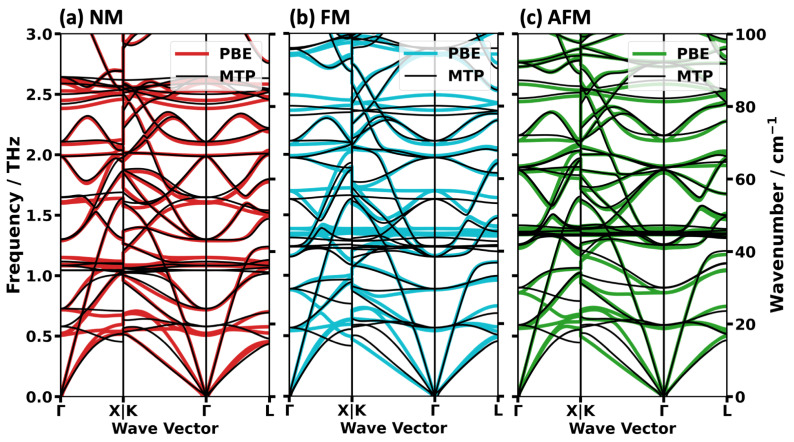
Comparison of low-frequency DFT/PBE and MTP-calculated phonon band structures for the NM (**a**), the FM (**b**) and the AFM (**c**) spin configurations. The DFT/PBE results are shown with thick colored lines and the predictions made by the spin-dependent MTPs are added on top using thinner black lines.

**Figure 9 ijms-25-03023-f009:**
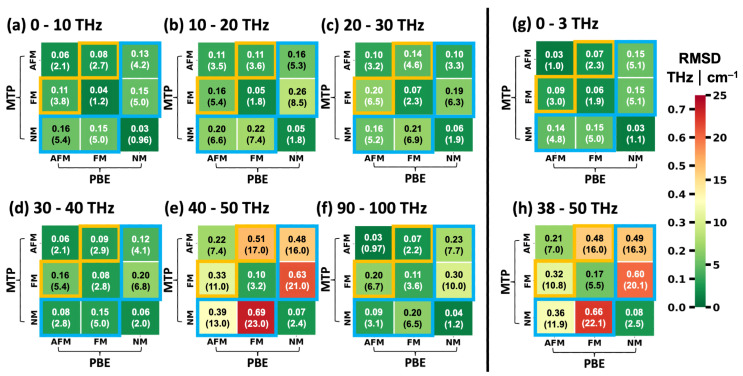
RMSDs between frequencies (wavenumbers) of Γ-point vibrations of HKUST-1 calculated using DFT/PBE (enforcing certain spin configurations) and calculated with spin-sensitively parametrized MTPs for frequency windows each covering 10 THz (panels (**a**–**f**)). Values are given in THz and also in cm^−1^ in parentheses. DFT-calculated unit cells have been used. Panels (**g**,**h**) illustrate the results for the low-frequency region and for a frequency range between 38 THz and 50 THz. For the columns, the DFT/PBE spin configurations were kept constant for the comparisons and for the lines that applies to the spin-dependent MTPs. The equivalence of specific vibrations is determined from the dot products of the eigenvectors [52]. The comparisons between FM and AFM states are highlighted by the orange squares and those between the NM configuration and the two spin-polarized ones by the blue rectangles.

**Figure 10 ijms-25-03023-f010:**
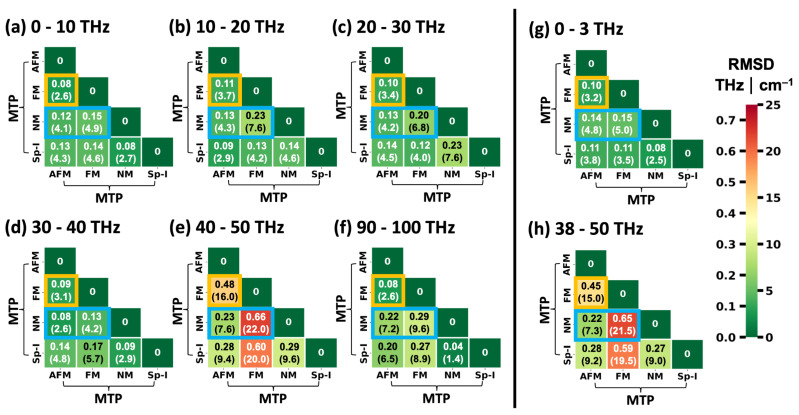
RMSDs between frequencies (wavenumbers) of Γ-point vibrations of HKUST-1 calculated using differently parametrized MTPs for frequency windows each covering 10 THz (panels (**a**–**f**)). Values are given in THz and also in cm^−1^ in parentheses. DFT-calculated unit cells have been used. Panels (**g**,**h**) illustrate the results for the low-frequency region and for a frequency range between 38 THz and 50 THz. The comparisons between FM and AFM states are highlighted by the orange squares and those between the NM configuration and the two spin-polarized ones by the blue rectangles.

**Figure 11 ijms-25-03023-f011:**
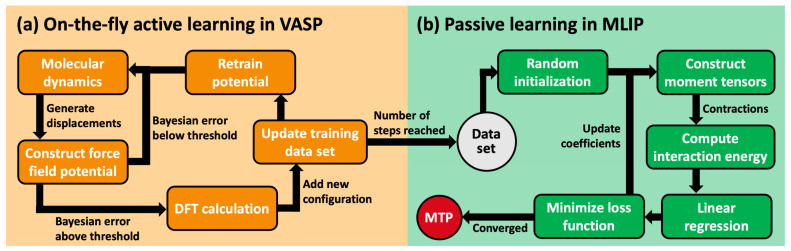
Workflow diagrams of the on-the-fly active-learning approach implemented in VASP [44] (**a**) and the passive learning approach in MLIP [40] (**b**).

**Table 1 ijms-25-03023-t001:** Relative energies per primitive unit cell (UC) of HKUST-1 containing six paddle wheels in different spin states and without spin polarization calculated using DFT/PBE. From the energies the exchange coupling constant for each paddle wheel (PW), J, can be determined. It is given in cm^−1^ to be consistent with literature, while it should be noted that for converting the value in cm^−1^ to an actual energy it must be multiplied with h and c.

E(AFM) [eV/UC]	0.00
E(FM) [eV/UC]	0.32
E(NM) [eV/UC]	0.92
J [cm^−1^/PW]	−215

**Table 2 ijms-25-03023-t002:** Local magnetic moments (M) of the atoms in the AFM and FM states of relaxed HKUST-1 structures computed with the PBE functional. In the AFM state, the sum of the moments of the entire unit cell and also per paddle wheel (M_total_) cancel. In contrast, for the FM state a value close to the expected value of 12 µ_B_ per unit cell is found.

Atom Indices for AFM State	M_AFM_ [μB]	Atom Indices for FM State	M_FM_ [μB]
**Cu1, Cu2**	±0.51	**Cu1**	0.55
**O1, O2**	±0.08	**O1**	0.09
**C1, C3, C4, C6**	0.00	**C1, C3**	0.00
**C2, C5**	0.00	**C2**	0.01
**H1, H2**	0.00	**H1**	0.00
**M_total_ [μB]**	0.00	**M_total_ [μB]**	11.6

**Table 3 ijms-25-03023-t003:** DFT/PBE-calculated group velocities v_g_ in the long-wavelength limit (i.e., sound velocities) in HKUST-1 for the NM, FM and AFM states. As can be concluded from Figure 4b, the highest group velocities are always obtained for the longitudinal mode (LA) independent of the direction of the wave vector. The group velocities for the two transverse acoustic (TA) modes in the Γ-L and Γ-K directions display very similar values. Further, we note that a group velocity of 1 THzÅ corresponds to 100 m/s. The values calculated using MTPs are given in round brackets.

	NM		FM		AFM	
	v_g_(TA) [THzÅ]	v_g_(LA) [THzÅ]	v_g_(TA) [THzÅ]	v_g_(LA) [THzÅ]	v_g_(TA) [THzÅ]	v_g_(LA) [THzÅ]
**Γ-X**	25.1 (23.6) 27.0 (23.4)	53.2 (53.1)	25.0 (23.6) 27.2 (23.2)	52.6 (53.4)	26.9 (24.1) 28.0 (23.6)	52.7 (53.6)
**Γ-K**	13.3 (13.0) 28.4 (24.3)	55.3 (54.9)	11.6 (13.3) 27.4 (23.9)	55.8 (55.5)	12.0 (14.2) 27.8 (24.5)	56.5 (55.6)
**Γ-L**	19.7 (16.8) 21.7 (17.5)	59.7 (58.2)	18.3 (16.7) 19.5 (18.0)	58.7 (57.9)	19.1 (17.6) 19.3 (18.3)	59.8 (58.3)

## Data Availability

The datasets generated in this work is available at the TU Graz Repository: https://doi.org/10.3217/4rwvv-n5j05 (accessed on 22 January 2024).

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
