# Peer review of "Predicting Spin-Dependent Phonon Band Structures of HKUST-1 Using Density Functional Theory and Machine-Learned Interatomic Potentials"

_ijms, 2024, doi:10.3390/ijms25053023_

Round 1
Reviewer 1 Report
Comments and Suggestions for Authors
Predicting Spin-Dependent Phonon Band Structures of HKUST-1: Density Functional Theory vs. Machine-Learned Interatomic Potentials
this manuscript is suitable for publication after English corrections and following corrections,
1. Title must be revised
2. introduction part is not more informative. Authors must revise it.
3. Why density functional theory is important? Authors must explain in introduction part.
Comments on the Quality of English LanguageFull manuscript must revise English corrections
Author Response
Reviewer 1:
Predicting Spin-Dependent Phonon Band Structures of HKUST-1: Density Functional Theory vs. Machine-Learned Interatomic Potentials
this manuscript is suitable for publication after English corrections and following corrections,
----------------
We thank the reviewer for the overall positive assessment of our manuscript. Regarding English corrections, we are not exactly sure what the reviewer is referring to. We have thoroughly read through the entire manuscript and did identify several instances in which there were typos and also negligent grammatical inconsistencies and missing words. We fixed these mistakes. Additionally, throughout the entire manuscript we clarified the discussion and shortened sentences. We had also asked a colleague, who is a native speaker and who has been a Solid State Physics Professor for many decades to read the manuscript. He agreed, but the extremely tight timeline that MDPI insisted on turned out to be inconsistent with his travel plans. Considering that EZ has published hundreds of papers, where there were never any problems with the English language, in view of the fact that he lived several years in the US, and considering that the other reviewers provided the maximum rating for the use of the English language in the manuscript, we are, however, confident that such an “external” proofreading would not be crucial.
----------------
- Title must be revised
----------------
Unfortunately, the brevity of this comment leaves us with little information on the specific reasons behind the suggested revision of the title.
We now changed the title to: “Predicting Spin-Dependent Phonon Band Structures of HKUST-1 Using Density Functional Theory and Machine-Learned Interatomic Potentials”
----------------
- introduction part is not more informative. Authors must revise it.
----------------
Unfortunately, the reviewer’s comment does not provide any information on what the problem with the introduction section of the manuscript might be.
As far as the current contents of the introduction is concerned, after a paragraph on the general purpose of MOFs we provided a very thorough and detailed discussion of HKUST-1, its spin properties and how these spin properties could be modelled reliably. We also describe the advantages of machine-learned force fields for efficiently modelling MOF properties. We do not see much that could be improved regarding these topics (save a small additional motivation for using DFT – see next comment by the reviewer). What has, however, indeed been missing so far is a short discussion on the relevance of phonons for all transport and thermodynamic properties of MOFs.
A paragraph on the relevance of phonons has now been added to the introduction section on page 1.
----------------
- Why density functional theory is important? Authors must explain in introduction part.
----------------
Quite generally, wave-function based methods have severe limitations, if one aims at modelling solids, as they are usually largely incompatible with employing periodic boundary conditions (although methods like CCSDT would provide an excellent description of correlation effects). Thus, they can only be applied to small clusters cut from the solids, possibly in combination with embedding methods. This causes severe limitations in the description of properties intimately linked to the quasi-infinite extent of crystalline materials. An example for such properties would be phonons as the quanta of lattice vibrations in extended solids. Considering that phonons are the main topic of the current manuscript, we are bound to employing periodic boundary conditions.
In contrast to the wave-function based methods, (semi-local) density functional theory is very well compatible with periodic boundary conditions and, thus, it is also the default ab initio methodology for describing phonons. This makes it the obvious ab initio method of choice for the current manuscript.
These aspects are now briefly described at the bottom of page 1/beginning of page 2.
As an additional remark, we also fixed an inconsistency in the discussion of the exchange coupling constant.

Reviewer 2 Report
Comments and Suggestions for Authors
The research addresses the issue concerning the spin-dependent vibrational properties of HKUST-1. The authors showed that the energy of the state with the antiferromagnetic (AFM) arrangement of spins is lower than that of the state with the ferromagnetic (FM) arrangement, thus corroborating an experimentally established fact and validating the appropriateness of the methodology used. Using the approved methodology, they carried out a systematic screening of the spin-state impact on phonon bands and densities of states in the various frequency regions for HKUST-1 and found that the acoustic phonon bands display very little dependence on the spin-state of the paddle-wheels, while some low-lying optical phonons manifest a strong dependence on the magnetic state of HKUST-1. As a main result, the authors conducted the parametrization of the moment tensor potential (MTP) with and without spin-sensitively parameters. Paying a special attention to the low-lying optical phonons, it was shown that the trained spin-including parameters MTP is capable of reproducing the spin-dependent variations in phonon properties of HKUST-1, while spin-ignorant MTP was unable to accurately predict the overall shapes of phonon density of states. This work demonstrates (on the example of HKUST-1) that the well-trained MTP may very well be used for approximating the results of highly demanding spin-DFT calculations, and encourages one to consider the machine-trained simulations as an efficient tool for the vibrational analysis of the solid-state structures.
The topic of the presented paper can be considered as an original one, and it is relevant to the special issue “Properties and Applications of Metal-Organic Frameworks.” This work fills a specific gap in the machine learning potentials for complex metal-organic frameworks with large unit cells such as HKUST-1. The manuscript is well-written, logically structured and can be interesting to a broad audience. The conclusions are consistent with the evidence and arguments presented in the text. The references are appropriate. The figures are in good quality. There are no amendments required for the Style or Grammer. Briefly, this is a well-prepared manuscript, which is ready for publication. And, I suggest to accept it after some minor corrections.
1. The manuscript is well detailed, which is very convenient for the reader. However, in my opinion, even more details should improve the clarity of the manuscript. This pertains the description of the MTP training. Machine-learning technology is the state-of-the-art approach, so there are many researches of the old-fashioned scientific manner who might experience difficulties in understanding the basics of the learning procedure. I suggest to extend the section 5.3 and add the training workflow diagram (logical diagram of the process).
2. In the connection to the first remark, I suggest to add some more references to reviews on the machine‑learned interatomic potentials. For example, the authors are advised to add the reference to the work of V. Eyert, et al. “Machine‑learned interatomic potentials: Recent developments and prospective applications,” J. Mat. Res., 2023, 5079. 10.1557/s43578-023-01239-8.
3. Lines 549-550: J = 215 and 185 cm-1. Should not there be the minus signs? The calculated J are negative (lines 138, 146).
Author Response
Reviewer 2:
The research addresses the issue concerning the spin-dependent vibrational properties of HKUST-1. The authors showed that the energy of the state with the antiferromagnetic (AFM) arrangement of spins is lower than that of the state with the ferromagnetic (FM) arrangement, thus corroborating an experimentally established fact and validating the appropriateness of the methodology used. Using the approved methodology, they carried out a systematic screening of the spin-state impact on phonon bands and densities of states in the various frequency regions for HKUST-1 and found that the acoustic phonon bands display very little dependence on the spin-state of the paddle-wheels, while some low-lying optical phonons manifest a strong dependence on the magnetic state of HKUST-1. As a main result, the authors conducted the parametrization of the moment tensor potential (MTP) with and without spin-sensitively parameters. Paying a special attention to the low-lying optical phonons, it was shown that the trained spin-including parameters MTP is capable of reproducing the spin-dependent variations in phonon properties of HKUST-1, while spin-ignorant MTP was unable to accurately predict the overall shapes of phonon density of states. This work demonstrates (on the example of HKUST-1) that the well-trained MTP may very well be used for approximating the results of highly demanding spin-DFT calculations, and encourages one to consider the machine-trained simulations as an efficient tool for the vibrational analysis of the solid-state structures.
The topic of the presented paper can be considered as an original one, and it is relevant to the special issue “Properties and Applications of Metal-Organic Frameworks.” This work fills a specific gap in the machine learning potentials for complex metal-organic frameworks with large unit cells such as HKUST-1. The manuscript is well-written, logically structured and can be interesting to a broad audience. The conclusions are consistent with the evidence and arguments presented in the text. The references are appropriate. The figures are in good quality. There are no amendments required for the Style or Grammer. Briefly, this is a well-prepared manuscript, which is ready for publication. And, I suggest to accept it after some minor corrections.
-------------
We thank the reviewer for the extremely positive assessment of our paper.
--------------
- The manuscript is well detailed, which is very convenient for the reader. However, in my opinion, even more details should improve the clarity of the manuscript. This pertains the description of the MTP training. Machine-learning technology is the state-of-the-art approach, so there are many researches of the old-fashioned scientific manner who might experience difficulties in understanding the basics of the learning procedure. I suggest to extend the section 5.3 and add the training workflow diagram (logical diagram of the process).
------------
We agree that illustrating the machine-learning process through a workflow diagram and expanding the explanation of the parametrization process can be beneficial. As we have utilized two distinct machine-learning routines – the on-the-fly active learning algorithm in VASP and the parametrization process of the MTPs – we have integrated the workflow of each of them into a combined diagram, establishing a clear connection between the two routines. The workflow diagram has been included now in Section 5.2, which deals with the generation of the training data sets in VASP. Moreover, we have expanded Sections 5.2 and 5.3 to provide thorough explanations of the new figure. We trust that this comprehensive presentation clarifies the entire process, making it more useful also for more traditionally inclined scientists.
Additionally, we now refer the readers to the extremely detailed description of the parametrization process we provided in a paper that came out earlier this year.
------------
- In the connection to the first remark, I suggest to add some more references to reviews on the machine‑learned interatomic potentials. For example, the authors are advised to add the reference to the work of V. Eyert, et al. “Machine‑learned interatomic potentials: Recent developments and prospective applications,” J. Mat. Res., 2023, 5079. 10.1557/s43578-023-01239-8.
----------------
We included the recommended reference and several additional references in line 101 of the revised manuscript.
----------------
- Lines 549-550: J = 215 and 185 cm-1. Should not there be the minus signs? The calculated J are negative (lines 138, 146).
----------------
We thank the reviewer for identifying this error! We have added the minus signs.
As an additional remark, we also fixed an inconsistency in the discussion of the exchange coupling constant.

Reviewer 3 Report
Comments and Suggestions for Authors
In their contribution “Predicting Spin-Dependent Phonon Band Structures of HKUST-1: Density Functional Theory vs. Machine-Learned Interatomic Potentials“, Strasser et al. report on two different approaches, which were employed to provide an insight into the vibrational properties of the aforementioned compound. In the framework of the research, all computations were accomplished within spin-polarized regimes, while state-of-the-art computational techniques have been employed. Furthermore, the research was carried out with great, while the contents presented in that remarkable work agree well with the scope of IJMS; yet, there are certain minor issues, which should be corrected prior to a publication of that excellent work:
- Under consideration of the forces acting in the inspected material, the authors made use of the DFT-D3 functional; however, I can remember that certain VASP-based computations accomplished by the Sauer Group in Berlin typically used a special approach to properly construct the potentials and functionals related to the large voids within such framework structures. Therefore, I am quite surprised that such an approach was not used by the authors for their computations. So, is it fully sufficient to describe correlation and exchange in the framework of the DFT-D3 method? By the way, the Grimme Group has recently released another approach (DFT-D4), which can simply be added to the VASP code and might also be helpful for further computations.
- In some cases, the functionals, which are pprovided for the d-bock elements, are not of utmost quality. Accordingly, it is possible the computationally determined bandgaps and magnetic moments differ from the experimentally observed data. Such problems could be solved by using a suitable Hubbard U parameter to properly depict correlation and exchange for the d-block elements. Hence, did the authors also encounter such problems for the copper-containing compound so that a Hubbard U parameter should be included?
- In the framework of their research, the authors computed the phonon band structures; however, it could also be possible to compute diverse spectra of the compound and compare them to the experimentally observed data from the literature. Such a comparison could also provide further hints, which approach works best.
- I am curious regarding the used computational resources. So, which of the two employed methods required a larger amount of computational resources?
- Please also include a DOI that is linked to the data repository.
Author Response
Reviewer 3
In their contribution “Predicting Spin-Dependent Phonon Band Structures of HKUST-1: Density Functional Theory vs. Machine-Learned Interatomic Potentials“, Strasser et al. report on two different approaches, which were employed to provide an insight into the vibrational properties of the aforementioned compound. In the framework of the research, all computations were accomplished within spin-polarized regimes, while state-of-the-art computational techniques have been employed. Furthermore, the research was carried out with great, while the contents presented in that remarkable work agree well with the scope of IJMS; yet, there are certain minor issues, which should be corrected prior to a publication of that excellent work:
-----------
We thank the reviewer for the very positive assessment of our work!
-----------
- Under consideration of the forces acting in the inspected material, the authors made use of the DFT-D3 functional; however, I can remember that certain VASP-based computations accomplished by the Sauer Group in Berlin typically used a special approach to properly construct the potentials and functionals related to the large voids within such framework structures. Therefore, I am quite surprised that such an approach was not used by the authors for their computations.
------------
We suppose that the reviewer is referring to the embedding approach described in C. Tuma and J. Sauer, Phys. Chem. Chem. Phys. 2006, 8, 3955-3965 (DOI: 0.1039/B608262A), T. Kerber, M. Sierka and J. Sauer, Journal of Computational Chemistry 2008, 29, 2088-2097 (DOI: 10.1002/jcc.21069) and J. Sauer, Accounts of Chemical Research 2019, 52, 3502-3510 (DOI: 10.1021/acs.accounts.9b00506). To our understanding, this approach has been primarily developed for describing molecules bonded to the pore walls via van der Waals interactions. Such guest molecules are not considered here. While the approached based on embedded MP2 calculations is highly sophisticated, a lot of time has passed since its development (when it was primarily compared to the D2 approach). Thus, it is not clear to us, to what extent it is still sufficiently superior to much more straightforward to use modern van der Waals corrections like the D3 and D4 approaches by Grimme et al. or the Many Body Dispersion (MBD) approaches by Tkatchenko et al.. A systematic comparison would certainly be worthwhile, but clearly goes beyond the scope of the current manuscript, which dose not deal with van der Waals interactions.
In fact, we have shown in a series of studies for molecular crystals that D3 and MBD-type van der Waals corrections (in contrast to the D2 or the TS approach) provide an exceptionally accurate description of vibrational and phonon properties of van-der Waals bonded molecular crystals compared to Raman scattering and inelastic neutron scattering experiments:
N. Bedoya-Martinez, B. Schrode, A. O. F. Jones, T. Salzillo, C. Ruzie, N. Demitri, Y. H. Geerts, E. Venuti, R. G. Della Valle, E. Zojer and R. Resel, The Journal of Physical Chemistry Letters 2017, 8, 3690-3695 (DOI: 10.1021/acs.jpclett.7b01634).
- Bedoya-Martinez, A. Giunchi, T. Salzillo, E. Venuti, R. G. Della Valle and E. Zojer, Journal of Chemical Theory and Computation 2018, 14, 4380-4390 (DOI: 10.1021/acs.jctc.8b00484).
T. Kamencek, S. Wieser, H. Kojima, N. Bedoya-Martinez, J. P. Dürholt, R. Schmid and E. Zojer, Journal of Chemical Theory and Computation 2020, 16, 2716-2735 (DOI: 10.1021/acs.jctc.0c00119).
Finally, it is not clear, how well the approach by Sauer et al. can be interfaced with the active learning strategy of VASP. The latter is, however, crucial for the current paper.
In view of the multiple arguments provided above, we do not consider it to be crucial to test the approach by Sauer et al. in the context of the current manuscript.
-------------
So, is it fully sufficient to describe correlation and exchange in the framework of the DFT-D3 method?
-------------
In view of the excellent performance of this van der Waals correction for describing phonons in molecular crystals (see above), we would definitely consider the DFT-D3 approach to be sufficient in the present context.
Nevertheless, to better assess the situation, we now added to the Supplementary Materials a rather extended test of how different van der Waals corrections and also different DFT functionals change the phonon properties of HKUST-1. This comparison shows that the main deviations between the different spin conformations of HKUST-1 prevail in all tested approaches even though the exact frequency values somewhat differ between the D3 and the MBD approach. There is, however virtually no difference between D3 and D4-based simulations
---------------
By the way, the Grimme Group has recently released another approach (DFT-D4), which can simply be added to the VASP code and might also be helpful for further computations.
--------------
We thank the reviewer for pointing out that the D4 correction of Grimme has been integrated into VASP (version 6.4.2). The bulk of the simulations for this manuscript has been performed using VASP (version 6.3.0), which lacked this D4 correction.
Following the suggestion by the reviewer, we now compiled VASP (version 6.4.2) with the external package ddft4 to test the impact of the D4 correction on the results of our study. Consequently, data generated with the D4 approach are now included in the above-mentioned comparison. As mentioned before, there are only very minor deviations between the D3 and D4 results.
---------
- In some cases, the functionals, which are provided for the d-bock elements, are not of utmost quality. Accordingly, it is possible the computationally determined bandgaps and magnetic moments differ from the experimentally observed data. Such problems could be solved by using a suitable Hubbard U parameter to properly depict correlation and exchange for the d-block elements. Hence, did the authors also encounter such problems for the copper-containing compound so that a Hubbard U parameter should be included?
---------
We do agree that in certain cases an explicit consideration the Hubbard U parameter can be useful for describing d- and f-block elements. At the same time, the approach is sometimes also criticized for the fact that the actual value of U is not always straightforward to determine independently. In fact, when discussing the issue with a colleague focusing on the simulation of highly correlated systems we had to learn that the ambiguity is even worse when correcting for the double-counting problem such that at some point the calculations entirely loose their predictive power. In view of our observation that even without considering U we obtained results for the exchange energy in excellent agreement with experiments, we refrained from applying the DFT+U approach here. Moreover, to the best of our knowledge combining DFT+U with the active learning approach used for parametrizing the force fields is also not documented. These aspects are now discussed in the beginning of the Materials and Methods section.
In fact, it has been discussed in literature that hybrid functional calculations can be well superior to the DFT+U approach. Thus, we also performed simulations using the PBE0 functional. The obtained results are discussed in detail in the Supplementary Materials. Due to high complexity of the studied material in combination with the extreme increase in computational costs when combining hybrid functional calculations with periodic boundary conditions, we, however, had to do these simulations with a rather small basis set. As this basis set is presumably not well converged, the PBE0 results must be considered as preliminary. This is also explicitly discussed in the paper.
--------------
- In the framework of their research, the authors computed the phonon band structures; however, it could also be possible to compute diverse spectra of the compound and compare them to the experimentally observed data from the literature. Such a comparison could also provide further hints, which approach works best.
-------------
In chapter S8 of the Supplementary Materials, we now discuss, how the IR spectra depend on the use van der Waals correction and in section S9 the impact of the functional is addressed. Finally, section S10 contains a comparison between theory and experiments, where one sees that the quality of the available experimental spectra is not sufficient for differentiate between different spin configurations. As far as the simulations are concerned, a further complication arises from the dependence of the peak positions on the used functional.
-------------
- I am curious regarding the used computational resources. So, which of the two employed methods required a larger amount of computational resources?
-------------
We had tried to indicate that already in the original version of the manuscript in the last paragraph of the conclusion section. Overall, we typically observe that if one was only interested in phonon band structures, the computational costs are similar with the exact ratio depending on a variety of aspects. This is now described more explicitly in the said paragraph. We now also much more explicitly describe, how the computational tasks “explode” as soon as other phonon-related properties (like heat transport or thermal expansion) are considered It is also discussed that none of these tasks stands any chance of being completed with ab initio methods especially for a system as complex as HKUST-1.
-------------
- Please also include a DOI that is linked to the data repository.
------------
We had not included a DOI in the original manuscript, as we expected that additional simulations would be necessary for the revised version of the manuscript. Now that these simulations have been added, a DOI has been generated, which has been included into the manuscript.
As an additional remark, we also fixed an inconsistency in the discussion of the exchange coupling constant.
